# Photon-Counting Computed Tomography in Atherosclerotic Plaque Characterization

**DOI:** 10.3390/diagnostics14111065

**Published:** 2024-05-21

**Authors:** Riccardo Cau, Luca Saba, Antonella Balestrieri, Antonella Meloni, Lorenzo Mannelli, Ludovico La Grutta, Eduardo Bossone, Cesare Mantini, Carola Politi, Jasjit S. Suri, Carlo Cavaliere, Bruna Punzo, Erica Maffei, Filippo Cademartiri

**Affiliations:** 1Department of Radiology, Azienda Ospedaliero Universitaria (A.O.U.) di Cagliari-Polo di Monserrato, S.S. 554, 09045 Monserrato, Italy; riccardocau00@gmail.com (R.C.); antonellabalestrieri@hotmail.com (A.B.); politi@unica.it (C.P.); 2Department of Radiology, Fondazione Monasterio/CNR, 56124 Pisa, Italy; antonella.meloni@ftgm.it (A.M.); filippocademartiri@gmail.com (F.C.); 3Department of Bioengineering, Fondazione Monasterio/CNR, 56124 Pisa, Italy; 4Department of Radiology, IRCCS SynLab-SDN, 80131 Naples, Italy; mannellilorenzo@yahoo.it (L.M.); carlo.cavaliere@synlab.it (C.C.); bruna.punzo@synlab.it (B.P.); emaffei@ftgm.it (E.M.); 5Department of Radiology, University Hospital “P. Giaccone”, 90127 Palermo, Italy; lagruttaludovico@gmail.com; 6Cardiology Unit, University of Campania Luigi Vanvitelli, 80138 Naples, Italy; 7Department of Neuroscience, Imaging and Clinical Sciences, “G.d’Annunzio” University, 66100 Chieti, Italy; cesare.mantini@gmail.com; 8Stroke Monitoring and Diagnostic Division, AtheroPoint™, Roseville, CA 95661, USA; jsuri@comcast.net

**Keywords:** photon-counting detector, carotid, coronary, atherosclerosis, plaque vulnerability

## Abstract

Atherosclerotic plaque buildup in the coronary and carotid arteries is pivotal in the onset of acute myocardial infarctions or cerebrovascular events, leading to heightened levels of illness and death. Atherosclerosis is a complex and multistep disease, beginning with the deposition of low-density lipoproteins in the arterial intima and culminating in plaque rupture. Modern technology favors non-invasive imaging techniques to assess atherosclerotic plaque and offer insights beyond mere artery stenosis. Among these, computed tomography stands out for its widespread clinical adoption and is prized for its speed and accessibility. Nonetheless, some limitations persist. The introduction of photon-counting computed tomography (PCCT), with its multi-energy capabilities, enhanced spatial resolution, and superior soft tissue contrast with minimal electronic noise, brings significant advantages to carotid and coronary artery imaging, enabling a more comprehensive examination of atherosclerotic plaque composition. This narrative review aims to provide a comprehensive overview of the main concepts related to PCCT. Additionally, we aim to explore the existing literature on the clinical application of PCCT in assessing atherosclerotic plaque. Finally, we will examine the advantages and limitations of this recently introduced technology.

## 1. Introduction

Cardiovascular disease is a leading cause of disability and premature mortality worldwide [1]. Although cardiovascular diseases are traditionally considered a disease of high-income countries, the greatest burden currently resides in low- and middle-income countries [2,3]. The process of atherosclerosis can start as early as the second decade of life and advance gradually over many years before causing symptoms or a cardiovascular event [4,5]. This extended period of disease development provides an opportunity to identify individuals in the early stages of atherosclerosis who could benefit from proactive preventive measures. The degree of stenosis has long been considered the only feature for risk stratification and guiding therapeutic decisions. Histopathological and imaging-based studies have identified plaque structure and composition as crucial determinants of either plaque vulnerability or stability, prompting researchers and clinicians to look beyond the degree of stenosis [6,7,8,9,10,11,12]. Presently, these plaque vulnerability features can be examined in vivo, thanks to advancements in non-invasive diagnostic techniques. Particularly, computed tomography (CT) stands out as a powerful non-invasive modality for evaluating the atherosclerotic process in terms of both luminal stenosis and plaque composition and morphology [8,9]. However, CT is still constrained by intrinsic limitations, including limited spatial resolution, suboptimal contrast resolution, and a contrast-to-noise ratio, as well as issues like blooming and beam-hardening artifacts (especially in patients with severe calcifications and/or stents), alongside suboptimal tissue characterization capabilities [13,14].

Photon-counting computed tomography (PCCT) represents a newly introduced detector technology in the realm of CT scanning. PCCT relies on a cutting-edge generation of X-ray detectors composed of semiconductor materials (e.g., cadmium telluride, cadmium zinc telluride, or silicon). These detectors directly convert individual X-ray photons into electron-hole pairs, enabling the precise detection and categorization of photons based on specific energy levels. This process effectively enables higher spatial resolution, reducing electronic noise with intrinsic spectral information [14,15,16,17] (Table 1).

Therefore, this technology promises to improve cardiovascular imaging in terms of atherosclerotic plaque evaluation, offering several benefits compared to conventional CT imaging and overcoming some of its limitations. This narrative review aims to provide a comprehensive overview of the main concepts related to PCCT. Additionally, our objective is to delve into the current body of the literature regarding the clinical utilization of PCCT for evaluating atherosclerotic plaque. Lastly, we will analyze the strengths and weaknesses inherent in this newly introduced technology.

Table 1 highlights the main differences in detector technology between photon-counting and conventional CT.

## 2. Photon-Counting Technology and Advantages in Plaque Evaluation

PCCT employs semiconductor layers, such as cadmium telluride, cadmium zinc telluride, or silicon, and utilizes a direct conversion technique that directly converts X-ray photons into electrical charges, unlike conventional CT systems that follow a two-step conversion process. Incident X-rays generate electron-hole pairs in the semiconductor, which are then separated and moved toward anodes by an electric field. As the electrons reach the anodes, they produce short current pulses, subsequently converted into voltage pulses. Importantly, the pulse height is directly proportional to the energy of the detected photon, enabling PCCT to provide energy information for each detected photon through its output signal. These energy data enable the precise categorization of photons into different energy bins, typically ranging from two to eight. To effectively reduce noise in the final signal, the lower threshold is usually set higher than the electronic noise level. Additionally, PCCT utilizes multiple electronic comparators and counters to process the output signal, determining the quantity of interacting X-ray photons by counting the generated pulses [17,18,19,20]. The primary clinical application of the new PCCT technology in atherosclerosis is closely associated with its ability to provide high-resolution visualization and detailed information about plaque characteristics thanks to the multi-energetic nature of the information collected.

The newly introduced PCCT technology has enhanced spatial resolution by eliminating reflectors or dead areas between pixels, enabling smaller pixel sizes with pitches as low as 0.15–0.225 mm at the isocenter, resulting in higher spatial resolution [21,22,23]. This enhanced spatial resolution allows for a more detailed visualization of plaque morphology and composition with increased contrast resolution, assisting in the better identification of high-risk plaque features. Additionally, PCCT technology enables quantifying both the total number of photons detected and their distribution across different energy levels. Indeed, in conventional CT images, the weighting of photons based on their energy tends to favor high-energy photons, potentially reducing the contrast-to-noise ratio by underweighting low-energy photons. Conversely, PCCT assigns equal weight to all photons, allowing for higher contrast, particularly for materials with low X-ray attenuation [15,24,25]. The energy-discriminating capability of PCCT enables the distinction of the electronic noise impact. By setting a threshold above the noise level (typically around 20–25 keV), PCCT filters out irrelevant electronic noise. The elimination of electronic noise using PCCT is advantageous in atherosclerosis plaque evaluation, reducing streak artifacts and improving signal uniformity [26,27]. Additionally, PCCT enables dose-efficient imaging, maintaining image quality while using lower radiation doses, ultimately enhancing diagnostic accuracy and patient safety in CT imaging [28]. PCCT has the capability to enhance the effectiveness of iodinated contrast media. It is a well-established fact that the linear attenuation coefficient of iodine increases as the X-ray energy decreases. This physical phenomenon presents an opportunity to use a lower quantity of intravenous contrast media while still achieving comparable diagnostic outcomes for a standard full-dose CT examination. The potential reduction of iodine administration in vessel evaluation was demonstrated by Emrich et al. using virtual monoenergetic image reconstruction at 40 KeV on PCCT, achieving a 50% reduction in the contrast media concentration [29] (Figure 1).

Beyond the reduction of iodine dosage, PCCT enables spectral separation through the simultaneous capture of CT data in different energy bins, differentiating various materials based on their distinct K-edge characteristics within a single image acquisition [30,31,32]. In the realm of atherosclerosis plaques, this PCCT capability enables discrimination between contrast medium-filled vessel lumens and calcified plaques based on their different K-edge characteristics, leading to a more comprehensive assessment of plaque components. However, there is a minimal difference in spectral attenuation between iodine-based contrast media and calcium. Therefore, several phantom and ex vivo studies have demonstrated the feasibility of other contrast agents with higher atomic numbers, such as holmium, hafnium, tungsten, and bismuth [23,33,34]. Additionally, atherosclerosis plaque evaluation may benefit from virtual monoenergetic image reconstruction at higher energies to reduce blooming artifacts from heavy calcifications. However, the attenuation of iodine decreases rapidly beyond 70 keV, currently limiting this approach. The introduction of new contrast media agents, such as tungsten, allows for the retention of high attenuation of up to 200 keV, enabling the visualization of vessel lumens and plaques beyond 70 keV [33]. The multi-energy capability of PCCT opens new horizons in the use of new contrast agents, even combined in a single acquisition to better define plaque composition and vulnerability. Si-Mohamed et al. investigated the PCCT spectral ability and its potential application in atherosclerotic plaques. In vitro experiments were initially performed to evaluate the capability of PCCT in differentiating gold, iodine, and calcium, followed by in vivo experiments using atherosclerotic rabbits with induced plaques. Gold nanoparticles were administered to seven rabbits with balloon injury-induced aortic atherosclerotic plaques on a high-cholesterol diet and four control rabbits on a normal diet, and PCCT imaging was conducted at various time points (5 min, 45 min, 1 day, and 2 days). Additionally, iodinated contrast-enhanced CT angiograms were also obtained. PCCT, utilizing multiple energy bins, effectively detected gold nanoparticles and distinguished them from calcifications and iodine, even when colocalized with calcium. An ex vivo histologic analysis confirmed the uptake of gold nanoparticles by macrophages within plaques, correlating with the concentrations measured in vivo. This correlation suggests that higher concentrations of gold nanoparticles detected by PCCT corresponded to an increased macrophage presence in advanced atherosclerotic plaques, indicating a significant advancement in molecular plaque characterization [23]. Similarly, nanoparticle-based contrast media were also evaluated in an animal study by Cormode et al., emphasizing the potential application of gold high-density lipoprotein nanoparticle contrast agents to identify the macrophage burden within the plaque and subsequently assess plaque vulnerability [30].

## 3. Features of Plaque Vulnerability

Histopathological and imaging-based studies have indicated that specific features of vulnerability within plaques are associated with acute coronary syndromes and cerebrovascular events [6,7,8,9,10,11,12]. Atherosclerosis is a multifaceted disease driven by a combination of genetic predispositions and environmental influences. Its progression unfolds through a series of intricate steps. It begins with the accumulation of low-density lipoproteins within the walls of arteries, initiating a cascade of events that include oxidative stress, immune system activation, and inflammatory responses. This initial insult leads to the formation of fatty streaks, early indicators of atherosclerosis, as intracellular lipids accumulate within the arterial intima. As the disease advances, inflammatory cells, predominantly macrophages and lymphocytes, accumulate at the site of injury. These immune cells adhere to the endothelial lining of blood vessels and subsequently infiltrate the arterial wall. This infiltration initiates the development of more complex lesions known as fibroatheromas, characterized by a lipid-rich necrotic core surrounded by a fibrous cap composed of connective tissue. The fibrous cap (FC) plays a crucial role in maintaining plaque stability. However, if the cap becomes thin or ruptures, it can lead to the exposure of the underlying plaque contents to the bloodstream, promoting thrombus formation. Additionally, an intraplaque hemorrhage, caused by the rupture of fragile vessels within the plaque, and angiogenesis, the formation of new blood vessels, further contribute to plaque destabilization [4,5,35,36,37]. Focal calcification is a common occurrence in atherosclerotic plaques, and its prevalence tends to increase with age [4,5]. The precise mechanisms underlying arterial calcification remain incompletely understood. Calcium deposition in blood vessels can occur in either the media or intima layers. Intimal calcification typically accompanies lipid buildup and inflammation, which are characteristic features of advancing atherosclerotic lesions. Conversely, medial calcification is more closely associated with advanced tissue degeneration [38]. Clinical evidence suggests that plaques responsible for myocardial infarction and/or cerebrovascular events often exhibit fewer calcifications [11,39]. Additionally, there are discernible differences in the patterns of plaque calcification. Currently, identifying features linked to a higher risk of plaque vulnerability is the focus of numerous pieces of non-invasive imaging research.

## 4. Clinical Application of PCCT in Carotid Plaque

The capability of PCCT to quantify the FC thickness, FC area, and lipid-rich necrotic core (LRNC) area was recently demonstrated in an ex vivo study by Dahal et al. on 20 excised plaques obtained from a carotid endarterectomy, with histopathological measurements serving as the ‘ground truth’. The FC thickness and area did not show a significant difference between the values obtained from the PCCT images and the corresponding histological images (*p* values > 0.05). The Bland–Altman plot confirmed these results, reporting mean differences for the FC thickness, FC area, and LRNC area measurements of 0.04 square root mm, −0.18 log10 mm^2^, and 0.10 log10 mm^2^, respectively [40]. Similarly, Shami et al. demonstrated that PCCT can detect features of plaque vulnerability. In their retrospective ex vivo study, the authors investigated advanced atherosclerotic plaques obtained during endarterectomies from the Carotid Plaque Imaging Project, comparing them with histological measurements. Plaque reconstructions were performed at different energy levels (45, 70, 120, and 190 keV). PCCT was able to distinguish among well-known features of plaque vulnerability, such as an intraplaque hemorrhage, as well as other plaque components including the FC, lipid, and necrosis [41]. Figure 2, Figure 3, Figure 4 and Figure 5 demonstrate some examples of carotid atherosclerotic plaques using PCCT.

Sartoretti et al. demonstrated the effectiveness of PCCT to detect and differentiate single photons based on their energies, enabling spectral separation in carotid atherosclerosis. The authors investigated an experimental tungsten-based contrast medium to enhance the visualization of carotid plaques compared to iodine-based contrast agents, utilizing histological specimens as a ‘ground truth’. Their findings showed that using tungsten as a contrast agent significantly reduced the image noise at both high and low radiation dose settings compared to iodine agents [34]. Similarly, the intrinsic spectral capability was reported by Healey et al., who demonstrated that PCCT could identify plaque vulnerability features such as LRNC, spotty calcification, and plaque ulceration in surgically obtained specimens from the carotid artery without the administration of contrast media, in comparison with histological imaging [42]. Another ex vivo study demonstrated that PCCT could distinguish the components of vulnerable atherosclerotic carotid plaque based on its characteristic energy-dependent attenuation characteristics related to differences in the photoelectric effect Compton scattering [43]. Another hallmark related to plaque development is the proliferation of vasa vasorum. Energy-integrating detector CT is limited in the evaluation of this biomarker of atherosclerosis due to the presence of blooming artifacts caused by contrast media in the lumen, which limits the assessment of subtle signals originating from the vasa vasorum within the arterial wall. PCCT benefits from enhanced spatial resolution, leading to reduced partial volume effects that contribute to blooming artifacts. Additionally, material decomposition algorithms enable accurate differentiation between high-density materials, such as iodine, and surrounding tissues, allowing for the detection of increased wall perfusion due to the proliferation of vasa vasorum [44,45,46].

## 5. Clinical Application of PCCT in Coronary Plaque

Si-Mohamed et al. demonstrated the superior image quality and diagnostic confidence of PCCT compared to an energy-integrating detector dual-layer CT. In their prospective study, the authors evaluated fourteen consecutive individuals who underwent both PCCT and energy-integrating detector dual-layer CT scans. They assessed the improvement in diagnostic quality scores for evaluating calcified and non-calcified plaques, as determined by three experienced cardiac radiologists. PCCT showed a 100% improvement in score for calcified plaques and a 45% improvement for non-calcified plaques. Additionally, the authors reported a 2.9-fold increase in the detectability index for non-calcified plaques in a phantom study [47]. An improved capability of quantitative coronary plaque characterization using PCCT has been reported in an in vivo study by Mergen et al. They compared ultra-high-resolution PCCT reconstructions (slice thickness of 0.2 mm) with the reference of standard reconstruction (slice thickness of 0.6 mm). Additionally, the authors evaluated different kernel reconstructions, including smooth (Bv40) and sharp (Bv64) vascular kernels. Their findings demonstrated that the volume of both fibrotic and lipid-rich plaque components was highest on ultra-high-resolution reconstructions due to reduced blooming artifacts, thereby optimizing risk stratification [48]. Similar results were also reported in an in vitro study by Rotzinger et al. under various conditions simulating patient sizes (small, medium, and large). The authors demonstrated that PCCT outperformed the energy-integrating detector CT for both non-calcified atherosclerotic plaque and lipid core detection tasks, regardless of plaque size, providing a superior detectability of 22–43% and 21–48%, respectively [49].

The spectral attenuation capability of PCCT to identify plaque components was reported by Boussel et al. The authors examined 23 plaques, comprising 10 calcified and 13 lipid-rich non-calcified samples obtained from postmortem human coronary arteries. Their study demonstrated PCCT’s ability to differentiate various plaque components, including lipid-rich cores and calcifications, from the normal artery wall and surrounding tissues by analyzing differences in the spectral attenuation [50]. Another ex vivo study demonstrated that PCCT was able to identify well-known markers of coronary plaque vulnerability, namely, plaque inflammation and spotty calcifications through material decomposition techniques in the presence of two contrasting materials such as iodine and gold [51]. The identification of calcification, especially when it is smaller than a few millimeters (e.g., spotty calcifications), a well-known marker of coronary plaque vulnerability, is challenging using conventional energy-integrating detector CT due to blooming artifacts. VanMeter et al. explored the effectiveness of PCCT in quantifying the calcium volume compared to conventional energy-integrating detector CT. They demonstrated a reduction in calcium blooming artifacts and improvements in calcification volume quantification accuracy thanks to improved spatial resolution through the utilization of smaller-sized detector elements [52]. Figure 6 and Figure 7 demonstrate some examples of coronary atherosclerotic plaques using PCCT.

Table 2 summarizes previous studies about the clinical application of PCCT in carotid and coronary plaque evaluation.

## 6. Potential Application of PCCT in the Light of Plaque-RADS and CAD-RADS

The Coronary Artery Disease Reporting and Data System (CAD-RADS) score [53] and its updated version [54] aim to stratify the risk of patients with coronary artery disease, primarily focusing on the degree of luminal stenosis and supplemented with ancillary features of high-risk plaque, including spotty calcification, low-attenuation plaque, positive remodeling, and a positive napkin-ring sign. The report should describe the identification of two or more of these high-risk features. Indeed, the presence of high-risk features impacts patient management independently of stenosis severity, often leading to more aggressive preventive therapy or the necessity of further non-invasive or invasive diagnostic tests [54]. In this scenario, PCCT enables a more comprehensive assessment of features of high-risk coronary plaque, allowing for an increased diagnostic performance in quantifying luminal stenosis, low-attenuation plaques (even in the presence of heavy calcification), and spotty calcifications. Figure 8. Therefore, this cutting-edge technology promises to better stratify patients, reducing unnecessary invasive procedures (in cases of luminal degree overestimation) and providing a more accurate assessment of plaque components and morphology [55]. The superior image quality and diagnostic confidence undoubtedly provide significant benefits in the clinical application of the carotid plaque-RADS [56]. This recently introduced scoring system is standardized and cross-modality designed for comprehensive reporting on carotid plaque composition and morphology, ranging from Plaque-RADS 1 to Plaque-RADS 4 based on specific features. However, CT currently has limitations in assessing certain aspects of plaque vulnerability, such as fibrous cap thickness/rupture (due to the limited spatial resolution and artifacts like the blue edge and halo effect) and the presence of an intraplaque hemorrhage (due to overlapping Hounsfield Unit values between lipid, fibrous, and hemorrhage components). Specifically, fibrous cap characteristics allow for discrimination between plaque-RADS 3a (thick fibrous cap), plaque-RADS 3b (thin fibrous cap), and plaque-RADS 4b (ruptured fibrous cap). On the other hand, the differentiation of plaque components into a lipid-rich necrotic core and intraplaque hemorrhage enables the discrimination between plaque-RADS 3a and plaque-RADS 4a, respectively. The plaque-RADS’s classification suggests that whenever practitioners encounter findings that do not definitively exclude the possibility of a relevant score upgrade, they should consider further examination. For example, identifying a Plaque-RADS score of 3a on the CT may prompt an additional investigation with magnetic resonance imaging (MRI) to rule out the presence of an intraplaque hemorrhage, which would upgrade the score to 4a. However, an MRI requires a longer acquisition time and it is less available in clinical practice in comparison to CT [56]. Therefore, the potential application of PCCT in defining fibrous cap characteristics, due to its improved spatial resolution and ability to discriminate plaque components through spectral capability, should aid physicians in clinical practice.

## 7. Clinical Perspective

The recent introduction of PCCT, thanks to its technological advancement, could undoubtedly offer advantages in routine clinical practice for both cerebrovascular and cardiovascular diseases. The improved spatial resolution enables the enhanced visualization of arteries and plaques and delineates complex anatomies. With the better delineation of the vessel lumen even in heavily calcified plaques, it allows for a more accurate assessment of the stenosis degree, consequently improving patient decision making. The superior image quality of PCCT enables a more comprehensive assessment of plaque imaging with the better identification of high-risk plaque features in both carotid and coronary arteries. In this scenario, PCCT represents a well-suited non-invasive imaging modality in the current paradigm shift from stenosis to plaque vulnerability, limiting further expensive non-invasive or invasive examinations. Additionally, this emerging technology could make CT examinations safer and available for vulnerable patients, including pregnant women, children, and patients with kidney disease, thanks to reduced radiation and contrast media exposure.

## 8. Limitations

However, certain intrinsic limitations, typical of all emerging technologies, must be considered to fully unlock the potential of PCCT in clinical practice. These technical challenges include phenomena like charge sharing, pixel crosstalk, and pulse pile-up. Charge sharing occurs when X-ray photons interact near pixel boundaries, causing the charge to spread across multiple adjacent pixel electrodes [57,58,59]. This can distort the distribution of detected photons and introduce artifacts in the image. Additionally, pixel crosstalk, such as K-fluorescence escape, further affects detector performance by limiting the practical pixel size in PCCT applications [60]. Pulse pile-up is another challenge, especially at very high X-ray flux rates, where overlapping voltage pulses from individual photon interactions can lead to errors in photon counting and energy measurements, compromising the image quality [61,62]. Another potential limitation in plaque assessment using PCCT is related to the utilization of different virtual monoenergetic images. Indeed, employing various energy levels for virtual monoenergetic images can result in significant changes in the Hounsfield unit values of the voxels, potentially influencing plaque volume estimations typically based on fixed Hounsfield unit thresholds. Vattay et al. demonstrated a significant decrease in plaque attenuation with increasing keV levels (from 723 ± 501 HU at 40 keV to 120 ± 112 HU at 180 keV), along with corresponding changes in plaque volume components. Particularly, the low-attenuation plaque volume increased with higher keV levels, suggesting that normalization protocols are crucial for ensuring comparable results across future studies in atherosclerosis plaque evaluation using PCCT [63]. Furthermore, PCCT’s use of alternative contrast agents, like nanoparticles, and higher doses of gadolinium compared to other imaging modalities raise concerns about patient safety and the optimal dosage. While promising, the clinical translation of nanoparticles is still experimental, requiring further research to evaluate their safety, efficacy, and potential advantages over conventional contrast agents. Despite its potential in cardiovascular imaging and other applications, further clinical validation and comparative studies are necessary to establish PCCT’s diagnostic accuracy, clinical utility, and impact on patient outcomes. Moreover, the high cost and limited availability of PCCT technology pose significant barriers to its widespread adoption in clinical settings. However, as the technology matures and becomes more established, a significant cost reduction is anticipated. Table 3 summarizes the benefits and limitations of photon-counting in atherosclerosis plaque evaluation.

## 9. Conclusions

The significant role of plaque morphology and composition in plaque vulnerability and stability has seen remarkable advancements in recent decades, with the introduction of PCCT emerging as a notable development in this field. PCCT offers numerous advantages over traditional CT technologies, including an improved spatial resolution and iodine signal, reduced electronic noise, lower radiation exposure, and a multi-energy capability. These advantages have demonstrated promising benefits in various aspects of carotid and coronary plaque evaluation, facilitating a more comprehensive assessment of imaging features related to plaque vulnerability. Further longitudinal multicenter studies are necessary to evaluate the clinical benefits of PCCT for cardiovascular imaging practice.

## Figures and Tables

**Figure 1 diagnostics-14-01065-f001:**
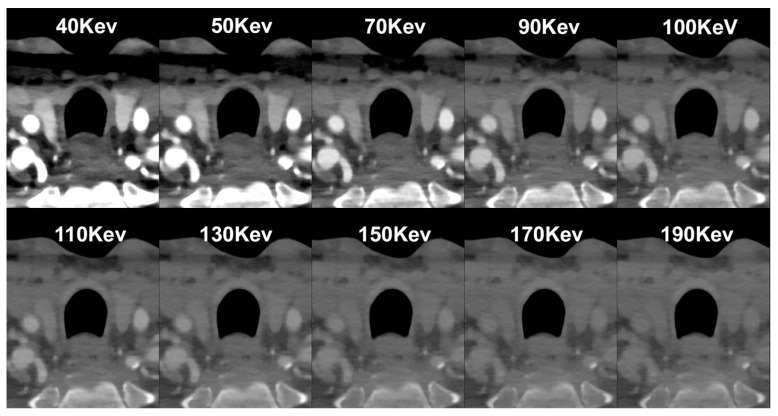
Spectral PCCT Angiography of the Carotid Arteries; spectral capabilities. The figure shows a sequence of axial scans at the same level of the neck (middle-distal Common Carotid Artery; CC) performed with Spectral PCCT Angiography of the Carotid Arteries. Each image is reconstructed with Monochromatic+ protocol, allowing a different settings of Kilo–electron–Volt (KeV), starting at 40 KeV up until 190 KeV. The scan was performed on a commercial whole-body Dual Source Photon-Counting CT scanner (NAEOTOM Alpha, Siemens Healthineers, Erlangen, Germany) with 0.2/0.4 mm slice thickness, 0.1/0.2 mm reconstruction increment, FOV 140–160 mm, resolution matrix of 512 × 512/1024 × 1024 pixels on the source axial reconstructions with a kernel filtering of Bv48–60 (vascular kernel medium-sharp), and with maximum intensity of Quantum Iterative Reconstruction (QIR 4). Abbreviations: PCCT = Photon-Counting CT; KeV = Kilo–electron–Volt; FOV = Field of View.

**Figure 2 diagnostics-14-01065-f002:**
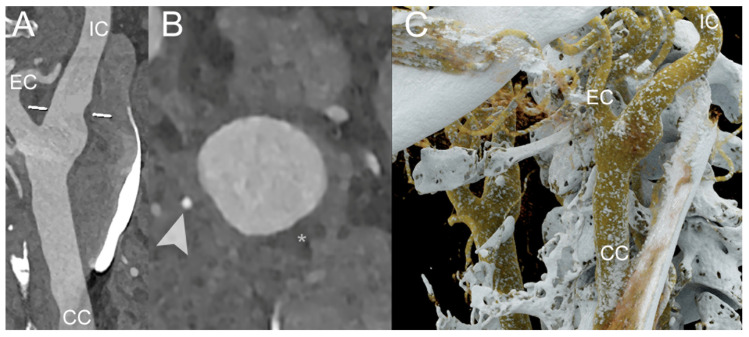
PCCT Angiography of the Carotid Arteries; mild atherosclerotic disease of extra-cranial left carotid system. The figure shows a PCCT Angiography of the Carotid Arteries and, in particular, a longitudinal multiplanar reconstruction ((**A**); MPR), an axial cross-sectional image orthogonal to the longitudinal axis of the vessel (**B**), and 3D Cinematic Rendering (**C**) of the extra-cranial left carotid artery system. The image shows only very mild non-calcified hypodense atherosclerosis (* in (**B**)) on internal carotid artery in proximity to carotid bifurcation with some nodular calcification (arrowhead in (**B**)). The scan was performed on a commercial whole-body Dual Source Photon-Counting CT scanner (NAEOTOM Alpha, Siemens Healthineers) with 0.2/0.4 mm slice thickness, 0.1/0.2 mm reconstruction increment, FOV 140–160 mm, and resolution matrix of 1024 × 1024 pixels on the source axial reconstructions with a kernel filtering of Bv48–60 (vascular kernel medium-sharp) and with maximum intensity of Quantum Iterative Reconstruction (QIR 4). The displayed image resolution is 100 microns. Abbreviations: PCCT = Photon-Counting CT; KeV = Kilo–electron–Volt; FOV = Field of View; CC = Common Carotid; IC = Internal Carotid; EC = External Carotid.

**Figure 3 diagnostics-14-01065-f003:**
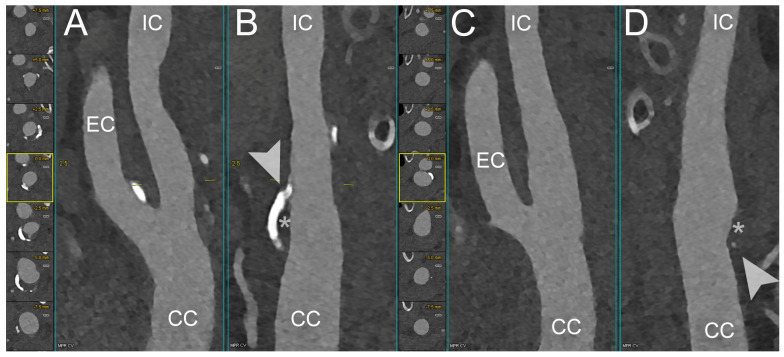
PCCT Angiography of the Carotid Arteries; mild bilateral extra-cranial atherosclerotic disease. The figure shows a PCCT Angiography of the Carotid Arteries and, in particular, orthogonal longitudinal multiplanar reconstruction of the right (**A**,**B**) and left (**C**,**D**) carotid bifurcation with corresponding series of axial cross-sections of the respective bifurcation on left of each one. The image shows mild mixed atherosclerosis on the right bifurcation with a calcified shell (arrowhead in (**B**)) and central hypodense core (* in (**B**)); instead, there is mild, predominantly non-calcified atherosclerosis (* in (**D**)) on the left bifurcation with a tiny nodular calcification (arrowhead in (**D**)). The scan was performed on a commercial whole-body Dual Source Photon-Counting CT scanner (NAEOTOM Alpha, Siemens Healthineers) with 0.2/0.4 mm slice thickness, 0.1/0.2 mm reconstruction increment, FOV 140–160 mm, and resolution matrix of 1024 × 1024 pixels on the source axial reconstructions with a kernel filtering of Bv48–60 (vascular kernel medium-sharp) and with maximum intensity of Quantum Iterative Reconstruction (QIR 4). The displayed image resolution is 100 microns. Abbreviations: PCCT = Photon-Counting CT; FOV = Field of View; CC = Common Carotid; IC = Internal Carotid; EC = External Carotid.

**Figure 4 diagnostics-14-01065-f004:**
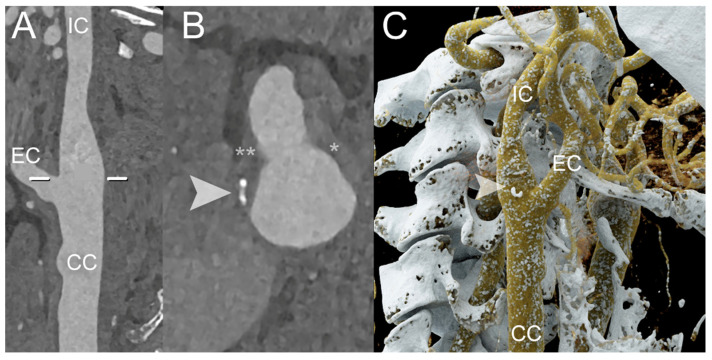
PCCT Angiography of the Carotid Arteries; mild extra-cranial atherosclerotic disease. The figure shows a PCCT Angiography of the Carotid Arteries and, in particular, longitudinal multiplanar reconstruction (**A**,**B**) and 3D Cinematic Volume Rendering of the right carotid bifurcation. The image shows mild mixed atherosclerosis on the right bifurcation with a small, calcified component (arrowhead in (**B**,**C**)), peripheral hyperdense non-calcified plaque (* in (**B**); i.e., more fibrotic/stable), and hypodense non-calcified plaque (** in (**B**); i.e., more lipidic/unstable). The scan was performed on a commercial whole-body Dual-Source Photon-Counting CT scanner (NAEOTOM Alpha, Siemens Healthineers) with 0.2/0.4 mm slice thickness, 0.1/0.2 mm reconstruction increment, FOV 140–160 mm, and resolution matrix of 1024 × 1024 pixels on the source axial reconstructions with a kernel filtering of Bv48–60 (vascular kernel medium-sharp) and with maximum intensity of Quantum Iterative Reconstruction (QIR 4). The displayed image resolution is 100 microns. Abbreviations: PCCT = Photon-Counting CT; FOV = Field of View; CC = Common Carotid; IC = Internal Carotid; EC = External Carotid.

**Figure 5 diagnostics-14-01065-f005:**
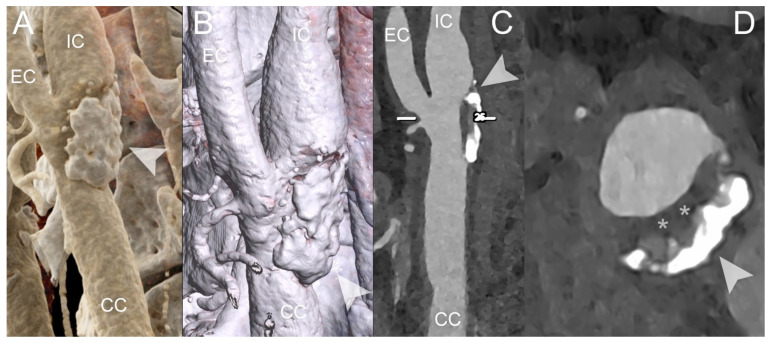
PCCT Angiography of the Carotid Arteries; severe extra-cranial atherosclerotic disease. The figure shows a PCCT Angiography of the Carotid Arteries and in particular 3D Cinematic Rendering (**A**), 3D Volume Rendering (**B**), longitudinal multiplanar reconstruction (**C**) and axial cross-sectional MPR along the longitudinal axis of the left carotid artery. The image shows severe mixed atherosclerosis predominantly calcified (arrowhead in (**A**–**D**)) associated with an inner hypodense non calcified plaque core on the intimal side (* in (**D**); i.e., more lipidic/unstable). The scan was performed on a commercial whole-body Dual Source Photon-Counting CT scanner (NAEOTOM Alpha, Siemens Healthineers) with 0.2/0.4 mm slice thickness, 0.1/0.2 mm reconstruction increment, FOV 140–160 mm, resolution matrix of 1024 × 1024 pixels on the source axial reconstructions with a kernel filtering of Bv48–60 (vascular kernel medium-sharp) and with maximum intensity of Quantum Iterative Reconstruction (QIR 4). The displayed image resolution is 100 microns. Abbreviations: PCCT = Photon-Counting CT; FOV = Field of View; CC = Common Carotid; IC = Internal Carotid; EC = External Carotid.

**Figure 6 diagnostics-14-01065-f006:**
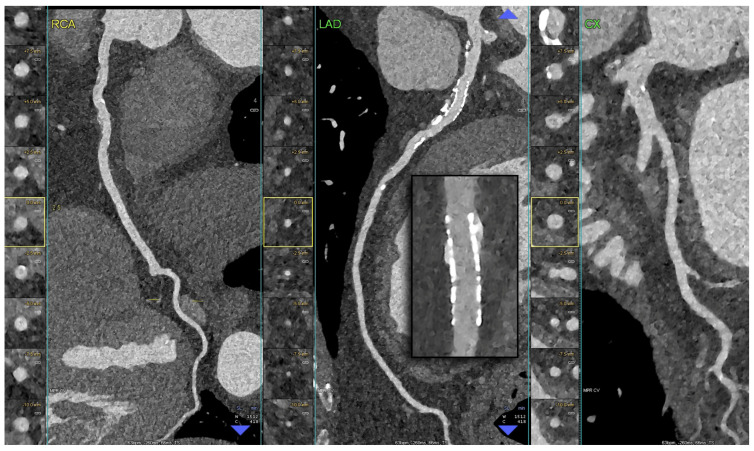
PCCT Angiography of the Coronary Arteries revealing diffuse atherosclerotic disease in both the left and right coronary arteries in a patient previously treated with PCI in the proximal left anterior descending artery. The figure depicts a PCCT Angiography of the coronary arteries, specifically a longitudinal multiplanar reconstruction and an axial cross-sectional image perpendicular to the longitudinal axis of each vessel (RCA, Right Coronary Artery—left panel; LAD, Left Anterior Descending—central panel; CX, Left Circumflex—right panel). The image illustrates diffuse-calcified coronary atherosclerosis, particularly in the left anterior descending artery. Furthermore, the stent is clearly visualized within its inner struts and lumen. The scan was conducted using a commercial whole-body Dual-Source Photon-Counting CT scanner (NAEOTOM Alpha, Siemens Healthineers) with a slice thickness of 0.2/0.4 mm, a reconstruction increment of 0.1/0.2 mm, FOV 140–160 mm, and a resolution matrix of 1024 × 1024 pixels for the source axial reconstructions. Kernel filtering was set to Bv60–72 (vascular kernel medium-sharp) and Quantum Iterative Reconstruction (QIR 4) was employed with a maximum intensity. The displayed image resolution is 100 microns.

**Figure 7 diagnostics-14-01065-f007:**
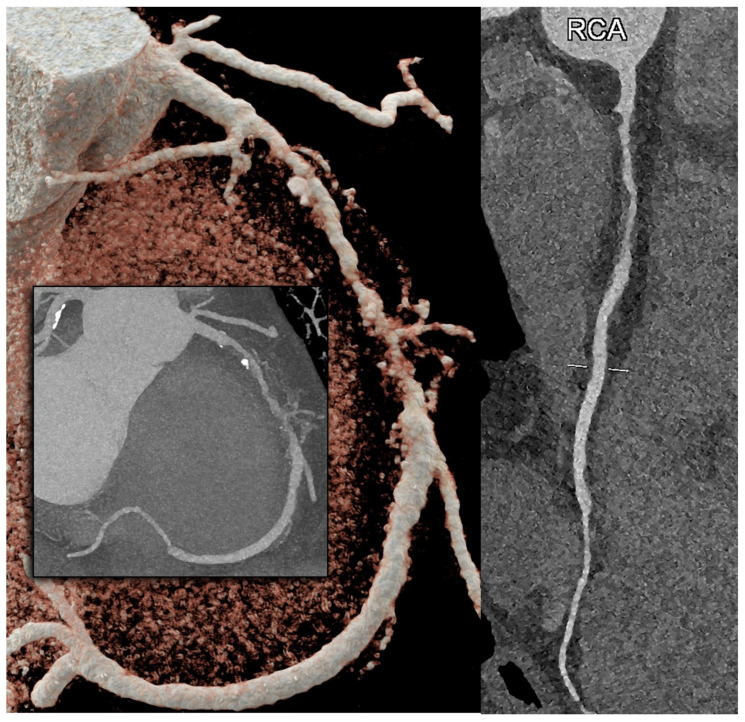
PCCT Angiography revealed obstructive non-calcified atherosclerotic plaque in the right coronary artery. The figure depicts a PCCT Angiography of the right coronary artery, specifically featuring a 3D cinematic rendering (left panel), a MIP (central figure of the left panel), and a longitudinal multiplanar reconstruction (right panel). The image illustrates a long, eccentric, moderate, obstructive, non-calcified coronary atherosclerosis in the proximal–middle segment of the right coronary artery. The scan was conducted using a commercial whole-body Dual-Source Photon-Counting CT scanner (NAEOTOM Alpha, Siemens Healthineers) with a slice thickness of 0.2/0.4 mm, a reconstruction increment of 0.1/0.2 mm, FOV 140–160 mm, and a resolution matrix of 1024 × 1024 pixels for the source axial reconstructions. Kernel filtering was set to Bv60–72 (vascular kernel medium-sharp) and Quantum Iterative Reconstruction (QIR 4) was employed with maximum intensity. The displayed image resolution is 100 microns.

**Figure 8 diagnostics-14-01065-f008:**
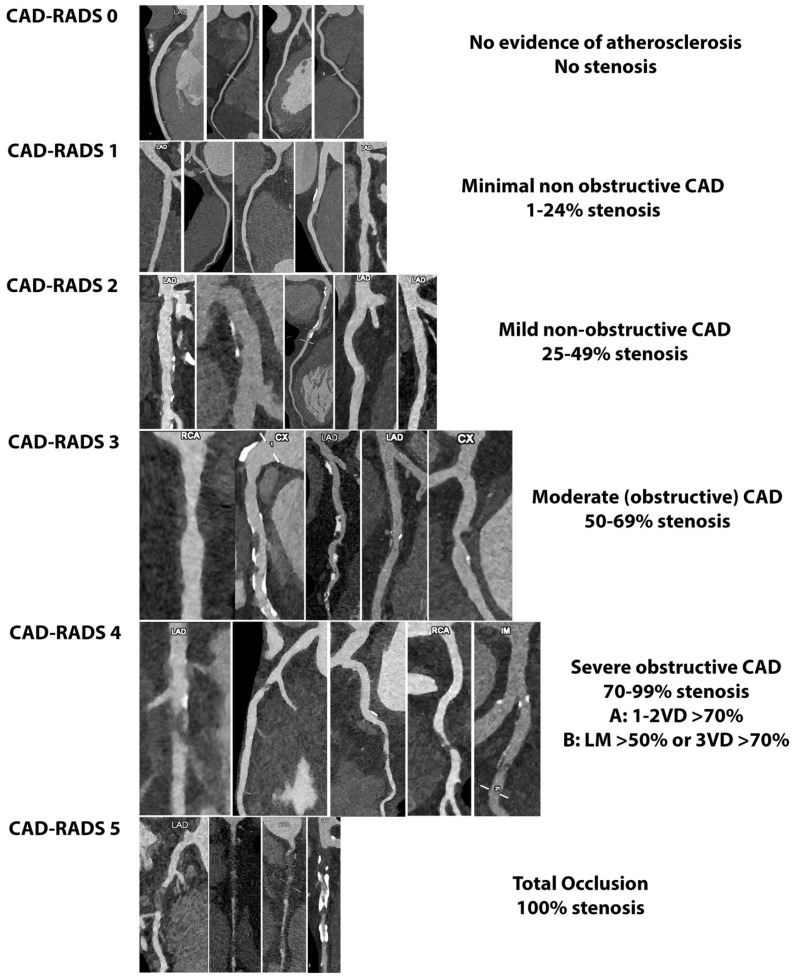
Practical examples of CAD-RADS categories using PCCT Angiography. The improved spatial resolution merged with artifact reduction enables a more comprehensive assessment of features of high-risk coronary plaque, allowing for increased diagnostic performance in quantifying luminal stenosis, low-attenuation plaques (even in the presence of heavy calcification), and spotty calcifications. The scans were conducted using a commercial whole-body Dual Source Photon-Counting CT scanner (NAEOTOM Alpha, Siemens Healthineers) with a slice thickness of 0.2/0.4 mm, a reconstruction increment of 0.1/0.2 mm, FOV 140–160 mm, and a resolution matrix of 1024 × 1024 pixels for the source axial reconstructions. Kernel filtering was set to Bv48–72 (vascular kernel medium-sharp) and Quantum Iterative Reconstruction (QIR 4) was employed with maximum intensity. The displayed image resolution is 100 microns.

**Table 1 diagnostics-14-01065-t001:** Main differences in detector technology between photon-counting and conventional CT.

	PCCT	Conventional CT
**Detectors**	Semi-conductive materials, such as cadmium telluride, cadmium zinc telluride, or silicon.	Energy-integrating detectors.
**Mechanism**	Direct conversion technology: X-ray photons are directly converted into an electrical signal. Each photon produces an electrical pulse, and the size (amplitude) of this pulse is directly proportional to the energy of the photon.	Indirect conversion technology: a scintillator layer first converts X-ray photons into visible light. This light is then captured by a photodiode layer and transformed into an electrical signal.
**Detectors**	Smaller detector element size without septa.	Detector element size affects spatial resolution.Thin septa between detectors are needed to prevent optical cross talk.
**Detector element size (mm^2^)**	0.225 × 0.225 mm^2^	0.3–0.7 mm^2^
**Resolution**	0.2 mm (high-resolution mode)0.4 mm (multi-energy mode)	0.6 mm
**Signal detection**	All photons are equally weighted regardless of their energy.	The detected signal represents the total energy of all photons, and high-energy photons tend to contribute more.
**Spectral characterization**	Intrinsic energy-dependent information with multi-energy acquisition.	Need for prospective selection of a dual-energy protocol (with the exception of the dual-layer detector).
**Advantages**	Improved spatial resolution. Decreased eletronic noise. Increased contrast. Artifact reduction. Enhanced capabilities for spectral imaging and material characterization.	Limited cost. Widespread availability.
**Disavantages**	High cost. Limited availability. Only one FDA-approved. PCCT.	Spatial resolution limited by detector element size. Measurement of total photon attenuation without information about each part of individual photon energy. Prospective spectral acquisition.

**Table 2 diagnostics-14-01065-t002:** An overview of previous studies on the clinical application of PCCT for evaluating carotid and coronary plaques.

Authors	Years	Model	Number of Patients	Clinical Application	Results
Dahal et al. [40]	2022	Ex vivo		Carotid	PCCT enables the assessment of fibrous cap thickness and area without significant difference in comparison with histological measurements
Shami et al. [41]	2024	Ex vivo		Carotid	PCCT was able to distinguish among well-known features of plaque vulnerability, such as intraplaque hemorrhage, as well as other plaque components including the fibrous cap, lipid, and necrosis
Sartoretti et al. [34]	2020	Ex vivo		Carotid	A multi-energy PCCT algorithm combined with tungsten-based contrast media which enables improved visualization of the vessel lumen and vessel wall with reduced image noise
Healy et al. [42]	2023	Ex vivo		Carotid	PCCT allows the identification of hallmarks of vulnerable plaque including a lipid-rich necrotic core, spotty calcifications, and ulcerations
Zainon et al. [43]	2012	Phantom		Carotid	PCCT demonstrated the capability to discriminate between plaque components, providing deeper insights into features associated with plaque instability
Marsh et al. [44]	2020	Phantom and in vivo (porcine)		Carotid	PCCT, using the blooming correction technique, enables the identification of injured artery’s vasa vasorum in comparison with control arterial walls
Rajendran et al. [45]	2017	Phantom and in vivo (porcine)		Carotid	PCCT, using the deconvolution technique allows the identification of increased vasa vasorum density in the wall compared to normal control artery
Marsh et al. [46]	2023	In vivo (porcine)		Carotid	PCCT is capable of detecting and quantifying the heightened perfusion observed within the carotid arterial wall due to an augmented density of vasa vasorum
Si-Mohamed et al. [47]	2022	In vivo (human)	14	Coronary	The proportions of score improvement observed with PCCT in comparison to EID-CT images were 100% for coronary calcification and 45% for noncalcified plaque
Mergen et al. [48]	2022	In vivo (human)	20	Coronary	Ultra-high-resolution scanning with PCCT improved visualization of fibrotic and lipid-rich plaque components
Rotzinger et al. [49]	2021	In vitro		Coronary	PCCT outperformed EID-CT images in detecting non-calcified plaques (AUC ≈ 95% vs. AUC ≈ 75%) and lipid-rich atherosclerotic plaques (AUC = 85% vs. AUC = 77.5%)
Boussel et al. [50]	2014	Ex vivo		Coronary	PCCT can identify plaque components by measuring differences in contrasting agent concentration and spectra attenuation
Baturin et al. [51]	2012	Phantom		Coronary	Multi-energy PCCT images using iodine and gold contrast agents enable the identification of markers of plaque vulnerability, namely spotty calcification and plaque inflammation
VanMeter et al. [52]	2022	Ex vivo		Coronary	PCCT demonstrated a reduction in calcium blooming artifacts and improvements in calcification volume quantification in comparison with EID-CT images

**Table 3 diagnostics-14-01065-t003:** Advantages and drawbacks of photon-counting for evaluating atherosclerotic plaques.

Benefits of PCCT	Limitations of PCCT
Higher spatial resolution	Technical issues (including charge sharing, pixel crosstalk, and pulse pile-up)
Electronic noise reduction	Lack of standardized protocol in virtual mono-energetic images
Improved iodine signal	High cost
Multi-energy acquisition	Limited availability
Artifact reduction	

## Data Availability

Data sharing is not applicable.

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
