# Peer review of "Photon-Counting Computed Tomography in Atherosclerotic Plaque Characterization"

_diagnostics, 2024, doi:10.3390/diagnostics14111065_

Round 1
Reviewer 1 Report
Comments and Suggestions for Authors
In this review, the authors assess photon-counting computed tomography for evaluating atherosclerotic plaque in coronary and carotid arteries. They highlight PCCT's superior imaging capabilities compared to traditional techniques, discuss its potential in assessing plaque composition and clinical applications, and address its limitations.
Congratulations to the authors for writing an interesting review on an innovative topic. However, a few minor aspects need to be considered:
1. Is it a narrative or systematic review? This should be specified both in the abstract and in the main body of the work.
- It could be beneficial for non-experts in the field to include a table illustrating the differences in the mechanisms, resolution, advantages, and disadvantages (including economic analysis) of PCCT compared to standard/conventional CT.
- Adding a clinical perspective to the work would be valuable, as would chronic coronary syndrome (for example, referring to paper 10.1016/j.ijcard.2022.07.038). There are indeed scenarios where CT is primarily indicated, while in others, it may not be optimal as a strategy (for instance, in patients with severe cardiovascular risk and severe symptoms, pregnancy, young women, patients with severe renal insufficiency, etc.). These aspects should be emphasized and elaborated upon in the text. Specifically, whether PCCT could improve or not the clinical approach to these patients should be evaluated.
- Lastly, some additional images of the coronary arteries would be useful.
Comments on the Quality of English Language
Minor English revisions and double-checking abbreviations are required.
Author Response
Is it a narrative or systematic review? This should be specified both in the abstract and in the main body of the work.
Dear reviewer, thank you for your comments. We have addressed this point in the revised version of the manuscript, both in the abstract and introduction
It could be beneficial for non-experts in the field to include a table illustrating the differences in the mechanisms, resolution, advantages, and disadvantages (including economic analysis) of PCCT compared to standard/conventional CT.
Dear reviewer, thank you for your comments. We have incorporated a new Table (Table 1) that illustrates the main difference in photon counting detectors technology in comparison with conventional CT. Additionally, the advantages and disadvantages of PCCT in atherosclerosis plaque evaluation are also reported in Table 3.
Adding a clinical perspective to the work would be valuable, as would chronic coronary syndrome (for example, referring to paper 10.1016/j.ijcard.2022.07.038). There are indeed scenarios where CT is primarily indicated, while in others, it may not be optimal as a strategy (for instance, in patients with severe cardiovascular risk and severe symptoms, pregnancy, young women, patients with severe renal insufficiency, etc.). These aspects should be emphasized and elaborated upon in the text. Specifically, whether PCCT could improve or not the clinical approach to these patients should be evaluated.
Thank you for pointing this out. We have added a new paragraph, namely "Clinical Perspective," that delineates the potential application of PCCT in the real-world clinical setting, as follows: The recent introduction of PCCT, thanks to its technological advancement, could undoubtedly offer advantages in routine clinical practice for both cerebrovascular and cardiovascular diseases. The improved spatial resolution enables enhanced visualization of arteries and plaques and delineates complex anatomies. With better delineation of vessel lumen even in heavily calcified plaques, it allows for a more accurate assessment of stenosis degree, consequently improving patient decision-making. The superior image quality of PCCT enables a more comprehensive assessment of plaque imaging with better identification of high-risk plaque features in both carotid and coronary arteries. In this scenario, PCCT represents a well-suited non-invasive imaging modality in the current paradigm-shift from stenosis to plaque vulnerability, limiting further expensive non-invasive or invasive examinations. Additionally, this emerging technology could make CT examinations safer and available for vulnerable patients, including pregnant women, children, and patients with kidney disease, thanks to reduced radiation and contrast media exposure.
Lastly, some additional images of the coronary arteries would be useful.
Dear reviewer, thank you for your comments. We have incorporated additional figures of the coronary arteries in the revised version of the manuscript (new figures 6, 7, and 8)
Reviewer 2 Report
Comments and Suggestions for Authors
The article provides detailed information on a relatively little-known subject.
Visual examples in the sections related to carotid plaques are satisfactory. Current sources were used. I have some minor suggestions:
1. More detailed general information about photon counting CT should be given in the introduction.
2. Case examples and images regarding coronary applications should also be presented.
3. Case examples regarding the use of CAD-RADS and PCCT should be given.
Author Response
- More detailed general information about photon counting CT should be given in the introduction.
Dear reviewer, thank you for your comments. We have incorporated a new Table (table 1), that summarizes the main differences between PCCT and conventional EID-CT.
- Case examples and images regarding coronary applications should also be presented.
Dear reviewer, thank you for your comments. We have incorporated additional figures of the coronary arteries in the revised version of the manuscript (new figures 6, 7, and 8)
- Case examples regarding the use of CAD-RADS and PCCT should be given.
Thank you for pointing this out. We have included practical examples of CAD-RADS categories using PCCT Angiography in the new Figure 8 in the revised version of the manuscript